# Alleviation of Collagen-Induced Arthritis by Crotonoside through Modulation of Dendritic Cell Differentiation and Activation

**DOI:** 10.3390/plants9111535

**Published:** 2020-11-10

**Authors:** Shih-Chao Lin, Chi-Chien Lin, Shiming Li, Wan-Yi Lin, Caitlin W. Lehman, Nicole R. Bracci, Sen-Wei Tsai

**Affiliations:** 1Bachelor Degree Program in Marine Biotechnology, National Taiwan Ocean University, Keelung 202301, Taiwan; sclin@mail.ntou.edu.tw; 2Institute of Biomedical Science, The iEGG and Animal Biotechnology Center, National Chung-Hsing University, Taichung 402204, Taiwan; lincc@email.nchu.edu.tw (C.-C.L.); an387usoa@smail.nchu.edu.tw (W.-Y.L.); 3Department of Medical Research, China Medical University Hospital, Taichung 404, Taiwan; 4Department of Medical Research, Taichung Veterans General Hospital, Taichung 40705, Taiwan; 5Department of Pharmacology, College of Medicine, Kaohsiung Medical University, Kaohsiung 80708, Taiwan; 6Hubei Key Laboratory for Processing and Application of Catalytic Materials, College of Chemistry and Chemical Engineering, Huanggang Normal University, Hubei 438000, China; Shiming@rutgers.edu; 7Department of Biomedical Sciences and Pathobiology, Virginia-Maryland College of Veterinary Medicine, Virginia Polytechnic Institute and State University, Blacksburg, VA 24063, USA; Woodsonc@vt.edu (C.W.L.); nbracci@vt.edu (N.R.B.); 8Department of Physical Medicine and Rehabilitation, Taichung Tzu Chi Hospital, Buddhist Tzu Chi Medical Foundation, Taichung 427, Taiwan; 9Department of Physical Medicine and Rehabilitation, School of Medicine, Tzu Chi University, Hualien 97004, Taiwan

**Keywords:** crotonoside, rheumatoid arthritis, dendritic cells, differentiation

## Abstract

Crotonoside, a guanosine analog originally isolated from *Croton tiglium*, is reported to be a potent tyrosine kinase inhibitor with immunosuppressive effects on immune cells. Due to its potential immunotherapeutic effects, we aimed to evaluate the anti-arthritic activity of crotonoside and explore its immunomodulatory properties in alleviating the severity of arthritic symptoms. To this end, we implemented the treatment of crotonoside on collagen-induced arthritic (CIA) DBA/1 mice and investigated its underlying mechanisms towards pathogenic dendritic cells (DCs). Our results suggest that crotonoside treatment remarkably improved clinical arthritic symptoms in this CIA mouse model as indicated by decreased pro-inflammatory cytokine production in the serum and suppressed expression of co-stimulatory molecules, CD40, CD80, and MHC class II, on CD11c^+^ DCs from the CIA mouse spleens. Additionally, crotonoside treatment significantly reduced the infiltration of CD11c^+^ DCs into the synovial tissues. Our in vitro study further demonstrated that bone marrow-derived DCs (BMDCs) exhibited lower yield in numbers and expressed lower levels of CD40, CD80, and MHC-II when incubated with crotonoside. Furthermore, LPS-stimulated mature DCs exhibited limited capability to prime antigen-specific CD4^+^ and T-cell proliferation, cytokine secretions, and co-stimulatory molecule expressions when treated with crotonoside. Our pioneer study highlights the immunotherapeutic role of crotonoside in the alleviation of the CIA via modulation of pathogenic DCs, thus creating possible applications of crotonoside as an immunosuppressive agent that could be utilized and further explored in treating autoimmune disorders in the future.

## 1. Introduction

Rheumatoid arthritis (RA) has been regarded as a chronic autoimmune disease, manifested by synovitis as well as cartilage and bone destruction. This disease results in disability in about 0.6 to 1% of the population in the United States [1]. The pathogenesis of RA is complicated, but it is generally recognized that over-reactive immune cells infiltrate synovial tissues, producing dysregulated cytokines and chemokines, leading to hyperplastic synoviocytes and activated osteoclasts that erode bone. The multiple signaling pathways associated with protein tyrosine kinases play a role in the activation or differentiation of cell types in various stages, rendering the initiation and progression of RA [2]. This includes the mitogen-activated protein kinase (MAPK) pathway, the Janus kinases (JAK) pathway, and the nuclear factor κ-light-chain enhancer of activated B cells (NF-κB) pathway. Despite inconclusive pathogenic factors, the autoantibodies such as anti-citrullinated protein antibodies are responsible for the majority of RA etiopathogenesis [3]. As a result, the aberrant autoimmunity associated with antigen-presenting cells (APCs), such as B cells and dendritic cells (DC), inevitably accounts for the initiation and progression of RA [4].

Owing to the extensive capability of DCs to activate naïve T cells, DCs are also regarded to play a crucial role in the pathogenesis of RA which has been thoroughly reviewed previously [5,6]. Also, the abundance of DCs found in synovial fluid and the aggravation of RA via direct administration of pro-inflammatory DCs in the joint supports that DCs could be associated with the onset as well as the progression of RA [7,8]. However, the interaction between DCs and RA is still not fully understood.

Crotonoside is the most abundant croton alkaloid extracted from seeds of *Croton tiglium L*. and active ingredients from *Croton tiglium* extract exhibits a wide variety of bioactivities that have been traditionally exploited for many indications, such as constipation, headache, abdominal and stomach pain, inflammation, and rheumatism and [9,10,11,12,13]. A more recent study revealed that seed extract suppressed the inflammatory responses elicited by LPS on microglia and astrocytes via inhibiting nitric oxide and tumor necrosis factor-α (TNF-α) and polarized microglia toward the M2 phenotype, potentially providing neuroprotectivie effects [14]. Additionally, crotonoside has been reported to be capable of suppressing acute myeloid leukemia via inhibiting Fms-like tyrosine kinase 3 (Flt3) and HDAC3/6 pathways [15]. Flt3 is a receptor tyrosine kinase and its ligand (Flt3L) a biomarker in the serum that predicts the progression of RA [16]. Flt3 is a soluble molecule that not only contributes to the progression but also the severity of RA pathogenesis [17]. The pathogenic role of Flt3L in aggravating RA was further confirmed by intra-articular administration of Flt3 inhibitor to methylated BSA-induced arthritic mice [18]. Due to the properties of-anti-inflammation and anti-Flt3-mediated pathway as well as the traditional rheumatism application of *Croton tiglium* extract, we postulated that crotonoside could be a potential bioactive agent for RA treatment and might have a strong association in the regulation of immune cells, particularly DCs, to exert their anti-arthritic effects. To this end, we conducted a detailed and comprehensive study to elucidate these mechanisms and evaluate the practical effects of modulating DC function by crotonoside in vitro and in collagen-induced arthritic (CIA) mice.

## 2. Material and Methods

### 2.1. Ethics Statement

All animal experimental protocols used in this study were conducted according to institutional guidelines and approved by the Institutional Animal Care and Utilization Committee of National Chung Hsing University, Taiwan (approval number NCHU-IACUC-108-002).

### 2.2. Animals

Eight-week-old female DBA/1 mice were obtained from the Jackson Laboratory, and OT-II TCR transgenic mice were provided by Dr. Clifford Lowell (UCSF, San Francisco, CA, USA). All mice were kept under controlled temperature, humidity, and light (12/12 h light-dark cycle) with water and food ad libitum.

### 2.3. CIA Induction and Crotonoside Treatment

CIA was performed as described previously with minor modifications [19]. Briefly, 2 mg/mL of bovine type II collagen (CII, Chondrex, Redmond, WA, USA) was dissolved in 10 mM of acetic acid and emulsified with a 1:1 ratio of complete Freund’s adjuvant (CFA, Chondrex). At the beginning of the experiments (day 0), the DBA/1 mice were intradermally injected with 0.2 mL emulsion at the tail base and CII emulsified 1:1 with incomplete Freund’s adjuvant (IFA; Chondrex) was injected as a booster on day 21. Crotonoside (Cat# CFN99524, ChemFaces, Wuhan, Hubei, China) was dissolved in a solvent of 10% DMSO and 90% glyceryl trioctanoate. The mice were randomly divided into four groups (*n* = 6 per group), including normal, CIA vehicle control, methotrexate (MTX, Cat# M9929, Sigma-Aldrich, Louis, MO, USA)-treated and crotonoside-treated mice at 25, 50, or 100 mg/kg. MTX is one of the most widely recognized disease-modifying anti-rheumatic drugs (DMARDs) as it is extensively prescribed to RA patients due to its effectiveness. It was originally designated as a chemotherapy agent for leukemia but has been repurposed to treat RA [20]. MTX (0.5 mg/kg) and crotonoside treatments were intraperitoneally administered once daily from day 21 to day 41 after the first immunization, whereas vehicle control mice received the solvent.

### 2.4. Scoring for Evaluating Arthritis Severity in CIA

Clinical symptoms of arthritis were evaluated visually in each limb and graded on a scale of 0–4 blindly by two independent operators using a semi-quantitative scoring system as described previously [19]. The maximal score for each limb was 4, with 0 representing no erythema or swelling and 4 indicating the most severe arthritic symptoms. The arthritis score for each mouse was the sum of all four limbs with a maximal score of 16. Paw volume was measured using a plethysmometer 37,140 (Ugo Basile SRL, Comerio, Italy).

### 2.5. Histological Assessment

On day 42, the mice were sacrificed by CO_2_ inhalation and joint tissues were removed from the hind paws followed by fixing with 4% paraformaldehyde, decalcifying in 5% formic acid, and embedding in paraffin. 5 μm thick of paraffin sections were stained with hematoxylin and eosin (H&E) or safranin O/fast green (Sigma, St. Louis, MO, USA) for visualization of tissue structure or proteoglycan content prior to microscopic evaluation. Histopathological changes, such as hyperplasia, cell infiltration, and cartilage destruction in synovial tissues, were blindly assessed by a pathologist and assigned scores of 0–4 as the previous described: 0, no changes; 1, mild changes; 2, moderate changes; 3, severe changes; 4, total destruction of architecture [21]. The CD11c expressions in the synovium were examined with immunohistochemistry staining that started with incubation of the primary rabbit anti-CD11c antibody overnight at 4 °C (1:200; Servicebio, Wuhan, China). Tissue sections were subsequently washed and incubated with HRP-conjugated goat anti-rabbit secondary antibody for 30 min at room temperature. Enzyme activity was then detected by adding 3,3′-diaminobenzidine (DAB) chromogen (LabVision Corp., Fremont, CA, USA). The quantification of infiltrated CD11c^+^ cells was performed through the determination of the integrative optical density (IOD) with ImageJ software.

### 2.6. Cytokine Measurement In Vitro and In Vivo

In vitro, bone marrow-derived DCs (BMDCs, 1 × 10^6^ cells/well) were pretreated with crotonoside for 4h followed by LPS (100 ng/mL) stimulation for 24 h. Levels of TNF-α, IL-1β, IL-6, IL-12p70, and IL-23 in the culture supernatants were determined by murine ELISA kits according to manufacturer instructions. Optical density (OD) values were measured with an ELISA reader at 450 nm (Tecan Sunrise). BMDCs were prepared from murine bone marrow as described previously with minor modifications [22]. Briefly, bone marrow cells were flushed out from the tibias and femurs of DBA/1 mice. Following lysis of red blood cells with ammonium chloride, the remaining cells were cultured in complete RPMI 1640 growth medium (HyClone, Chicago, IL, USA) supplemented with 10% heat-inactivated fetal bovine serum (FBS), 2 mM of L-glutamine, 100 U/mL of penicillin, 100 mg/mL of streptomycin, 0.05 mM of 2-mercaptoethanol, 10 mM HEPES (pH7.4) (all from Sigma), 20 ng/mL of recombinant mouse granulocyte-macrophage colony-stimulating factor (rmGM-CSF), and 10 ng/mL of rmIL-4 (PeproTech, Rocky Hill, NJ, USA). On day 0, 5 × 10^6^ cells per well were seeded in 6-well plates (3 mL/well) and incubated at 37 °C in 5% CO_2_. On days 2, 4, and 6, non-adherent cells were gently removed and fresh new complete medium supplemented with FBS, 20 ng/mL rmGM-CSF, and 10 ng/mL rmIL-4 were added. By day 7, the non-adherent and loosely adherent cells were collected and regarded as immature BMDCs.

In vivo, serum and paw tissues were obtained on day 42. Serum was obtained following centrifugation of mouse blood at 10,000× *g* for 10 min. 100 mg of frozen tissues were homogenized with 1 mL of tissue lysis buffer containing protease inhibitor cocktail (UltraCruz^®^, Santa Cruz Biotechnology, Santa Cruz, CA, USA) followed by determining total protein concentrations with a bicinchoninic acid protein assay kit (Thermo Fisher Scientific, Waltham, MA, USA). Levels of TNF-α, IL-6, IL-17A, IL-10 in both serum and paw homogenates were measured by using murine ELISA kits as per the manufacturer instructions.

### 2.7. CCK-8 Cell Viability Assay

1 × 10^6^ cells per well of bone marrow-derived DCs (BMDCs) were pretreated with 0–100 μM of crotonoside for 4 h followed by LPS (100 ng/mL) stimulation for 24 h. CCK8 reagent was added to each well, incubating for 4 h followed by optical density determination at 490 nm by a microplate reader (Tecan Sunrise). The mean optical density (OD, absorbance) from four wells of the indicated group was used to calculate the percentage of cell viability as the formula as follows: [(OD_experiment_ − OD_blank_)/(OD_control_ − OD_blank_)] × 100%, where OD_experiment_ is the absorbance of cells treated with either LPS + 0.1% DMSO or LPS + crotonoside-treated cells, and OD_blank_ is the absorbance of a well with medium only. OD_control_ is the absorbance of cells treated with 0.1% DMSO only.

### 2.8. Lactate Dehydrogenase (LDH) Assay

LDH cytotoxicity assay kit (Cayman Chemical) was employed to assess the cellular toxicity induced by crotonoside. Similar to CCK-8 assay, BMDCs were pretreated with crotonoside and stimulated by LPS followed by collecting the supernatant via centrifuging at 1200 rpm for 5 min. 100 µL/well of the supernatant was transferred to a new 96-well plate and an equal volume of LDH reagent was added to each well. After incubating for 30 min at room temperature, the concentration of LDH was measured with a microplate reader (TECAN) at a wavelength of 490 nm. The LDH release level (% of positive control) was recorded as the percentage of (OD_test_ − OD_blank_)/(OD_positive_ − OD_blank_), where OD_test_ is the optical density of cells treated with either 0.1% DMSO, LPS + 0.1% DMSO or LPS + crotonoside; OD_positive_ is the optical density of 1% Triton X-100-treated cells and OD_blank_ is the absorbance of a well with medium only.

### 2.9. Flow Cytometry Analysis of Surface Markers on DCs

For detection of BMDC differentiation in vitro, the 1:1000 dilution of anti-CD11c antibody was utilized to detect the expression of CD11c marker via an Accuri 5 flow cytometer following 50 and 100 μM of crotonoside were added to the culture media. Fresh crotonoside was supplemented each time whenever the medium was changed.

For BMDC maturation assay, DCs (1 × 10^6^ cells/well) were pretreated with a crotonoside (0–100 μM) for 4 h followed by stimulation of 100 ng/mL of LPS for 24 h. Expression of CD11c and co-stimulatory markers, CD40, CD80, and MHC class II were analyzed by Accuri 5 flow cytometer using 1:1000 dilution of FITC-labeled anti-mouse CD11c, PE-anti-mouse CD40, PE-anti-mouse CD80, PE anti-mouse I-A/I-E (MHC II) antibodies (BioLegend, San Diego, CA, USA) along with corresponding isotype-matched control antibodies. The mean fluorescence intensity (MFI) of CD40, CD80, and MHC class II was automatically estimated by BD Accuri C6 system software following gating with forward side scatter (FSC) and CD11c^+^ expression. A similar protocol was utilized for DC phenotyping experiments based on CD40, CD80, and MHC-II but no pretreatment of crotonoside.

### 2.10. Determination of Splenic T Cell Subtypes

Splenocytes were prepared at the end of the experiment, day 42, after treating with or without crotonoside and single-cell suspension was obtained using mechanical disruption and passed through a steel mesh (mesh size: 30 μm). 2 × 10^6^ cells/mL of splenocytes were cultured in 24-well plates along with complete medium and stimulated with 20 μg/mL of CII for 72 h. Brefeldin A (BioLegend) was added for the last 6 h during the stimulation. Activated splenocytes were stained for surface CD4 marker with 1:1000 of phycoerythrin (PE)-conjugated anti-mouse CD4 antibody (Biolegend) and fixed in fixation buffer (BioLegend) followed by intracellular staining with 1:1000 of FITC-conjugated anti-mouse IFN-γ and IL-17A antibodies (BioLegend). The percentages of T cell subsets were then analyzed by using an Accuri 5 flow cytometer.

### 2.11. Assay of Antigen-Specific T Cell Activation in Co-Culture Settings

To establish a co-culture system of T cells and DCs, immature BMDC were incubated with magnetic beads-conjugated anti-CD11c mAb (Miltenyi Biotec, Auburn, CA, USA) followed by positive selection through paramagnetic columns (LS columns; Miltenyi Biotec) according to the manufacturer instructions. Meanwhile, OVA_323–339_ peptide-specific CD4^+^ T cells were negatively purified from the splenocytes of OT-II mice by using EasySep Mouse CD4^+^ T Cell Isolation Kit (Stem Cell, Grenoble, France). The purity of both CD11c^+^ DCs and CD4^+^ T cells were confirmed to be over 90%. Purified DCs were pulsed with OVA_323–339_ (OT-II) peptides (2 or 10 μg/mL) for 6 h followed by addition of 100 ng/mL of LPS, LPS plus 0.1% DMSO, or LPS along with crotonoside for an additional 18 h and then transferred to culture with CD4^+^ T cells in a 1:5 ratio (DC: 5 × 10^4^ cells/well; T cell: 2.5 × 10^5^ cells/well) in U-bottomed 96-well microtiter culture plates. During the last 18 h of the 72 h culturing, cell proliferation was quantified based on radioactivity due to uptake of [^3^H]-thymidine (1 μL Ci/well; NEN-DuPont, Boston, MA, USA). After harvesting cells, liquid scintillation counting was used (Beckman Instruments, Palo Alto, CA, USA). Supernatants from the co-culture system were collected to measure the IFN-γ and IL-17A levels by ELISA (eBioscience) after 72 h of incubation.

### 2.12. Western Blot Analysis

Cell lysate of crotonoside-pretreated CD11c^+^ BMDCs were collected at 0, 30, 60, and 90 min lysed with RIPA lysis buffer, and determined the phosphorylation of MAPK pathways. The protein concentrations were determined using bicinchoninic acid (BCA) assay. 40 μg of the protein samples were boiled and loaded into 8–10% gradient SDS-PAGE gels and then electrotransferred to polyvinylidene difluoride (PVDF) membranes. The membranes were blocked with superblock T20 blocking buffer (Thermo, Rockford, IL) and were hybridized with 1:1000 dilution (1:5000 for GAPDH) of primary antibodies (Cell Signaling Technology) that recognize phospho-p38 (Thr180/Tyr182) (cat. No. 4631), p38 (cat. No. 8690), phospho-ERK (Thr202/Tyr204) (cat. No. 4370), total ERK (cat. No. 3192), phosphor-JNK (Thr183/Thr185) (cat. No. 4668), JNK (cat. No. 9252), and GAPDH (cat. No. 2118) at 4 °C overnight. After washing, membranes were incubated with HRP-conjugated mouse anti-rabbit secondary antibody (cat. No. 111-035-003; Jackson ImmunoResearch Laboratories) in 1:2000 dilution at 4 °C overnight prior to developing with ECL reagent (GE Healthcare Life Sciences), visualizing with the Hansor Luminescence Image System (Taichung, Taiwan). Densitometric analyses for target bands were performed with ImageJ software (National Institute of Health) and normalized against corresponding GAPDH and the time 0 DMSO vehicle control levels.

### 2.13. Statistical Analysis

Data were expressed as the mean ± standard deviation. Two-tailed Student’s *t*-test was applied to compare two individual data sets. One-way or two-way ANOVA with a post hoc Dunnett test was used to compare multiple experimental groups with GraphPad Prism v5.0 software (La Jolla, CA, USA). A *p*-value of less than 0.05 was considered a significant difference.

## 3. Results

### 3.1. Crotonoside Reduced Arthritic Severity in CIA Mice

We first examined the therapeutic effect of crotonoside on CIA DBA1/J mice via i.p. injection. As shown in Figure 1A–C, vehicle-treated control CIA mice had more severe erythema, paw swelling, and increased clinical arthritic scores compared to normal mice. In contrast, crotonoside-treated groups (50 and 100 mg/kg) and methotrexate treated mice (MTX, 0.5 mg/kg) [23], exhibited improvement of paw erythema and swelling as well as lower arthritis severity scores over the entire observation period without any significant changes in their body weights (Figure 1D).

When examining the hind knee joint tissue structures through H&E staining, we found that joint structures of MTX-, 50 (Cro-50), and 100 (Cro-100) mg/kg of crotonoside-treated mice resembled the normal knee joint tissues (Figure 2A). There was a remarkable reduction in the synovial hyperplasia and inflammatory cell infiltration in the 50 and 100 mg/kg of crotonoside-treated as well as in the MTX-treated control group (Figure 2A,C). Selected joint sections were stained with Safranin O to evaluate the content of proteoglycans. Safranin O proportionally stains the tissue and the staining can indicate if a joint structure is intact [24,25]. The vehicle-treated CIA group exhibited reduced proteoglycan content (Figure 2B), indicating articular cartilage was disrupted. Whereas crotonoside-and MTX-treated CIA mice were observed to be significantly ameliorated in proteoglycan depletion within the same area, supporting that crotonoside interrupted the progression of CIA.

### 3.2. Reduced Synovial DC Infiltration and Splenic DC Activation by Crotonoside

Following confirmation that crotonoside treatment alleviated the severity of CIA, we investigated whether DCs are involved in the alleviation of arthritis induced by crotonoside. Thus, we examined the distribution of CD11c-expressed cells in joint synovial tissues through immunohistochemistry staining with and without crotonoside treatment. Histological images in Figure 3A,B revealed that CD11c^+^ cells accumulated in the vehicle-treated CIA mouse synovial membrane tissues, but CIA mice that received crotonoside treatment exhibited fewer CD11c^+^ cells gathering in the same tissues.

Furthermore, we evaluated the activation levels of splenic DCs from CIA mice. Splenocytes were examined for the expression of costimulatory molecules, CD40, CD80, and MHC class II, as indications of DC activation. Their expression level in the cellular population were quantified by flow cytometry [26]. The flow cytometry data indicates an elevation of these surface markers in an arthritic setting. Crotonoside treatment was able to suppress splenic DC activation in CIA mice, reducing the levels to ones comparable to normal mice (Figure 3C and Appendix A). These data suggest that the dormancy activation of splenic DCs could be modulated by crotonoside in the CIA mouse model.

### 3.3. Proinflammatory Cytokines and Inflammatory T Cell Subsets Were Suppressed through Crotonoside Administration

To further clarify the microenvironment of mouse joint synovium and the anti-arthritic activity of crotonoside, we measured the levels of multiple proinflammatory cytokines by ELISA. The levels of TNF-α, IL-6, and IL-17A as well as the anti-inflammatory cytokine, IL-10, in both sera and paw tissues from CIA mice after crotonoside treatment were examined. The data in Figure 4 not only confirms that all four cytokines tested were substantially higher following the establishment of CIA systemically and locally, but more importantly it pinpointed that TNF-α, IL-6, and IL-17A were significantly stymied upon crotonoside treatment in the sera (Figure 4A) and paw homogenates (Figure 4B). Conversely, the anti-inflammatory IL-10 exhibited no statistically significant changes regardless of crotonoside dosage.

Owing to the trend of reductions in TNF-α and IL-17A and the pathogenic roles of Th1 and Th17 in RA [27], we further examined the number changes in T cell subsets of CD4^+^ IFN-γ^+^ (Th1) and CD4^+^ IL-17A^+^ (Th17). The splenocytes from CIA mice were isolated and stimulated with CII protein ex vivo for 72 h followed by measurement of Th1 or Th17 frequency via flow cytometry. In keeping with the cytokine results, we found that the proportions of Th1 (Figure 5A and Appendix A) and Th17 (Figure 5B and Appendix A) were back within normal ranges with crotonoside treatment despite the induction of CIA. These data suggest that the administration of crotonoside inhibited the local and systemic proinflammatory cytokine production as well as decreased the frequencies of Th1/Th17 that contribute to the retention of joint structures.

### 3.4. Crotonoside Inhibited BMDC Differentiation

We performed an in vitro study to elucidate whether DC differentiation was also affected by crotonoside as only a small amount of DCs can be retrieved from peripheral blood, synovial tissues, and spleens in mice [28]. To this end, we isolated myeloid cells from CIA murine bone marrow-derived cells and added crotonoside (0, 50, and 100 μM) into the culture medium during incubation and investigated the proportions of CD11c^+^ cells within the myeloid cells. Compared to the DMSO mock control, the percentages of CD11c^+^ cells were dose-dependently and significantly reduced upon the addition of crotonoside (Figure 6), suggesting that crotonoside constrained DC differentiation.

### 3.5. Crotonoside Inhibited Co-Stimulatory Molecules and Cytokine Production of BMDCs Induced by LPS

Following up on the suppression of DC differentiation in Figure 6, we attempted to characterize more changes in BMDCs affected by crotonoside. We activated and treated immature BMDCs from naïve mice with LPS and crotonoside, respectively followed by measurement of co-stimulatory molecule expressions. The levels of co-stimulatory marker expressions, CD40, CD80, and MHC-II, all of which enable DCs to prime downstream T cell differentiation, were repressed by crotonoside at both treatment concentrations (Figure 7).

Given the fact that DCs could contribute to arthritis by producing inflammatory cytokines and that these cytokines can guide naïve T cell differentiation [5,6], we explored the DC cytokine profile stimulated by LPS followed by treating with or without crotonoside. We observed that in addition to remarkably reducing the proinflammatory cytokines; TNF-α and IL-1β, the Th1-biased cytokine (IL-12) as well as Th17-biased cytokines (IL-6, IL-23) were also profoundly reduced as a result of crotonoside treatment (Appendix A). It is also notable that the 100 μM concentration of crotonoside did not result in significant cytotoxicity by CCK-8 and LDH assays (Appendix A).

### 3.6. Crotonoside Impaired the Ability of BMDCs to Activate Antigen-Specific T Cells

Lastly, we wanted to depict the interplay among crotonoside, DCs, and its downstream target, the T cells. To this end, we established a co-culture system where naïve CD4^+^ T cells were directly isolated from OT-II transgenic mice that had CD4 T cells specific to ovalbumin (OVA) [29], and cultured them with immature BMDCs preincubated with OVA. Prior to co-culturing, the immature BMDCs were activated with LPS and treated with crotonoside overnight followed by the evaluation of T cell proliferation and Th1/Th17 cytokines via thymidine incorporation assay and ELISA, respectively. The proliferation results in Figure 8A shows that the capability to proliferate antigen-specific T cells was substantially inhibited by crotonoside in comparison to T cells in the inflammatory scenario. The signature cytokines Th1 (IFN-γ) and Th17 (IL-17A) were also reduced upon crotonoside treatments (Figure 8B), arguing that the ability to present an antigen of mature DCs was suppressed, leading to less T cell proliferation and decreased cytokine production.

## 4. Discussion

Tyrosine kinases are involved in a number of cellular response pathways that lead to RA pathogenesis. For example, vascular endothelial growth factors (VEGFs) facilitate angiogenesis which contributes to a higher density of vascular distribution in synovia of RA patients [30]. In addition to VEGF, Flt3 ligand has been demonstrated to induce osteoclastogenesis and the differentiation of osteoclasts in the absence of macrophage colony-stimulating factor (M-CSF) [31], making inhibition of tyrosine kinases an attractive and applicative target to remedy RA. With this concept, we applied the Flt3 inhibitor, crotonoside, on CIA mice and demonstrated that the differentiation and activation of DCs, including the ability to produce cytokines, were remarkably reduced and subsequently impaired the Th1 and Th17-biased activation. All of which synergistically contribute to the severity of arthritis in mice as graphically summarized in Figure 9.

While the specificity of the interaction between crotonoside and Flt3 pathway blockage was not fully investigated in the current study, it has been reported that Flt3L could be a prominent target for RA therapy as shown in previous clinical research. Ramos et al. found that the Flt3L levels in serum and synovial fluid/tissues were profoundly higher in RA patients than healthy subjects or even gout patients [32]. Moreover, the Flt3L expressions on CD1c^+^ myeloid DCs and CD304^+^ plasmacytoid DCs (pDCs) isolated from RA synovial fluids were significantly higher than those in healthy individuals. In line with their findings, our results show that the development and activation of BMDCs were suppressed by crotonoside treatment and thereby reduce the immunopathological impacts in the CIA model. However, we did not look into the functional changes of monocyte or natural killer cells which were also observed by Ramos et al. with higher Flt3 expressions [32].

As the first study to exploit the inhibitory activity of crotonoside in RA, it is not a new concept to target Flt3 pathway and validate the potential of Flt3 inhibitor. The Flt3 ligand is crucial for bone marrow precursors to develop and differentiate into conventional DCs and pDCs [33], both of which could be associated with the inflammation of synovial tissues [34,35]. As such, the blockage of the Flt3 pathway is a reasonable approach in the alleviation of RA. In line with our study using crotonoside, other Flt3 inhibitors, such as sunitinib and sorafenib, approved by Food and Drug Administration (FDA) for chemotherapy, were also demonstrated to be capable of reducing arthritis induced by methylated BSA or type II collagen in murine models [2,17,36,37,38]. Of note, Dehlin et al. found that sunifinib treatment not only reduced the synovitis and bone destruction but also inhibited DC formation in the spleen and their antigen-presenting ability [18]. Likewise, in this study, we also observed decreased activations of splenic and bone marrow-derived DCs as well as the reduction APCs, cytokine production, and the priming downstream T cell differentiation, which support the aforementioned study.

Previously, little was known about the mechanism crotonoside exerts as a natural glycoside to achieve its various bioactivities even though it was first isolated back in 1932 [39]. It was not until recently that, Li et al. demonstrated that crotonoside inhibited the phosphorylation cascade of Flt3 pathway (Appendix A) and selectively suppressed the histone deacetylase 3 and 6 (HDAC3/6) in acute myeloid leukemia cells [15]. HDAC3 is known to play an important role in T cell development and maturation [40] which could be associated with the reduction of CD4 T subtypes we observed in Figure 6. Moreover, the HDAC3 gene knockdown by siRNA or treatment with HDAC3 inhibitor down-regulated the inflammatory cytokine and chemokine genes in RA fibroblast-like synoviocytes (FLS), particularly IL-1β and IL-6 (Figure 4 and Appendix A) [41]. As a result, crotonoside might also directly reduce the inflammation through FLS in the CIA model in addition to DCs.

The pharmacokinetics of crotonoside and the application of AML treatment in vivo have been investigated previously [22,42]. In the present animal study, we intraperitoneally administrated 25–100 mg/kg of crotonoside and observed a significant improvement of arthritis on CIA mice with at least a dosage of 50 mg/kg. Compared to similar studies, crotonoside was given to tumor xenograft NOD-SCID mice with up to 70 mg/kg intraperitoneally. Other Flt3 inhibitors, like sunitinib, were injected via i.p. route at 30 and 60 mg/kg, sorafenib was administered orally at 10 and 30 mg/kg, and SB1578 was provided to CIA mice at 70–210 mg/kg [36,37,43]. Therefore, the dosages of crotonoside we used in our study still fall within a reasonable range. However, more studies are needed to elucidate the pharmacodynamics of crotonoside and to determine an effective dosage as well as any off-target side effects crotonoside may have before proceeding to clinical studies.

## 5. Conclusions

Our study utilized crotonoside to explore as a remedy to treat CIA mice, revealing highly optimistic results. These results include fewer arthritic characteristics, a decrease in immune cell infiltration to synovial tissues, and decreased inert DC and T cell activations. Moreover, we comprehensively elucidated that the Th1 and Th17-biased cytokine production driven by DCs was subsequently stymied upon the administration of crotonoside. Our findings provide a more robust mechanistic view of the application of crotonoside in arthritic disorders.

## Figures and Tables

**Figure 1 plants-09-01535-f001:**
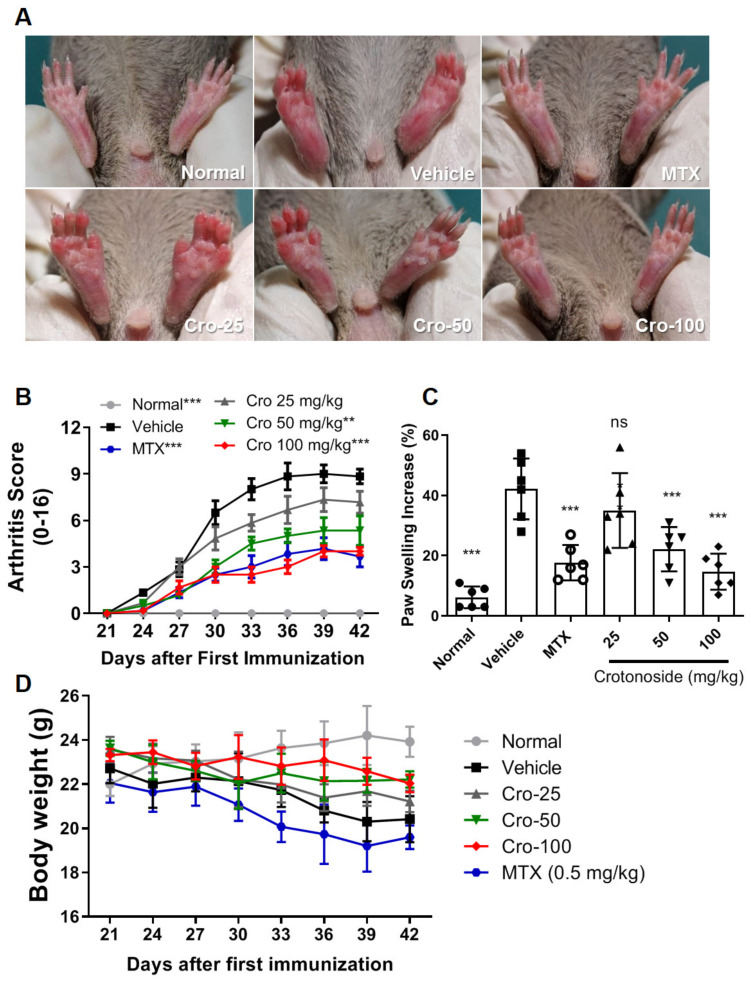
Crotonoside ameliorated the severity of collagen-induced arthritis (CIA) in mice. (**A**) Representative photographs of the hind paws of CIA mice on day 42. (**B**) Arthritis scores were monitored every 3 days after immunization booster. (**C**) Paw swelling levels were measured through the volumes of the hind paws using a paw volume plethysmometer. (**D**) Body weights of normal, vehicle-treated CIA, and CIA mice treated with methotrexate (MTX, 0.5 mg/kg) or crotonoside (25, 50, or 100 mg/kg); Data are representative of three independent experiments, and values are expressed as mean ± SD (*n* = 6 per group). ** *p* < 0.01 and *** *p* < 0.001 versus vehicle-treated CIA control mice calculated with two-way ANOVA.

**Figure 2 plants-09-01535-f002:**
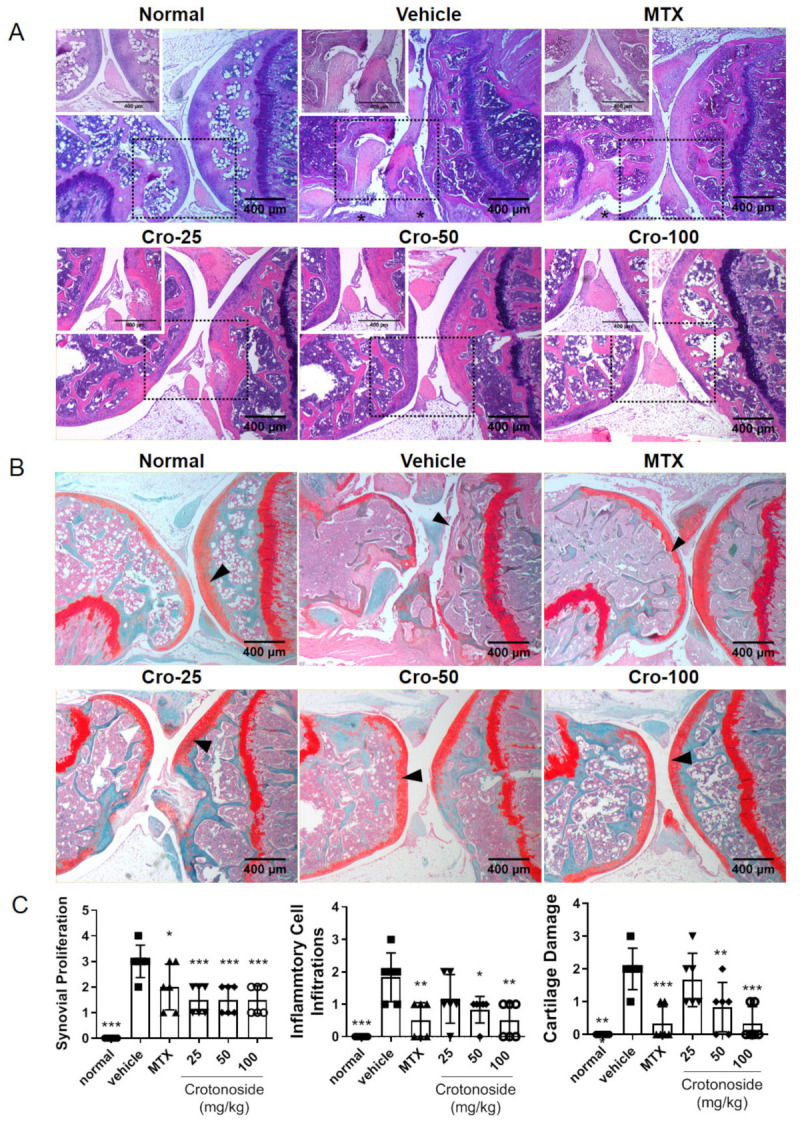
Histopathological damages associated with collagen-induced arthritis were mitigated by crotonoside. Hind knee joint tissue sections stained with (**A**) hematoxylin and eosin (H&E, magnification of background image, ×100; insert, ×200) or (**B**) Safranin O staining representation of different mouse groups. (**C**) Histopathological indications, synovial proliferation, cell infiltration, and cartilage damage were quantified based on 200× magnification images. Representative data are shown from one of three independent experiments and values are expressed as mean ± SD (*n* = 6 mice per group). Asterisks and arrowheads indicate the synovial hyperplasia and proteoglycans, respectively. * *p*  < 0.05, ** *p*  < 0.01, *** *p*  < 0.001, versus CIA vehicle group, as determined by a one-way ANOVA with Dunnett test.

**Figure 3 plants-09-01535-f003:**
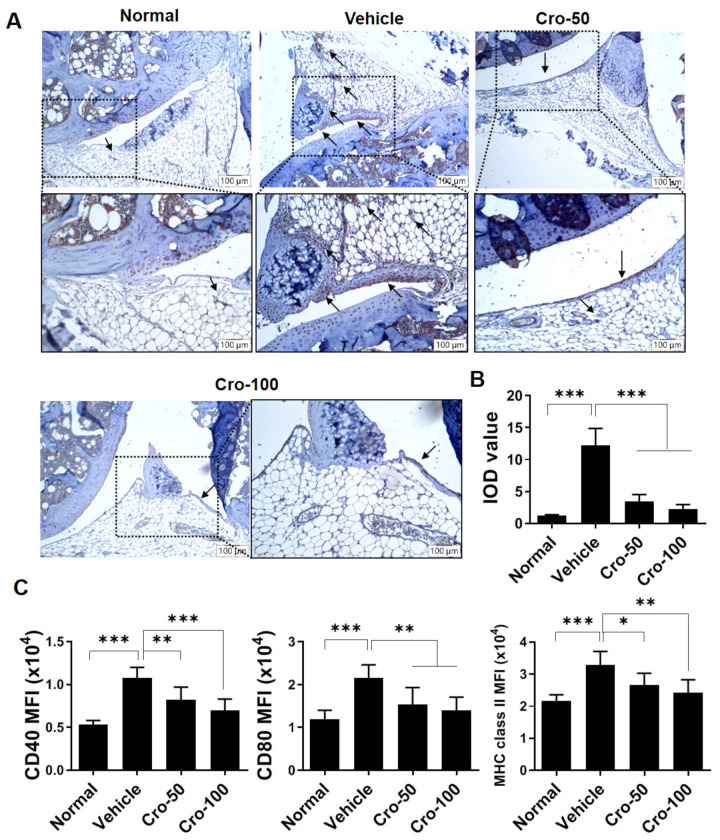
Infiltration or activation of either synovial or splenic CD11c^+^ cells were suppressed by crotonoside. (**A**) Detection of CD11c expressing cells (brown, indicated by arrows) by immunohistochemical analysis in the knee joint tissues. (**B**) The integrated optical density (IOD) values of CD11c^+^ cells in synovial membrane tissues were qualified and compared among the groups. (**C**) Mean fluorescence intensity (MFI) of CD40, CD80, and MHC class II expression was determined by gating CD11c^+^ DCs in normal and CIA mice with or without crotonoside treatments at indicated concentrations. Data (*n* = 6 per group) are from one of three experiments and expressed as the mean ± SD. * *p*  < 0.05, ** *p*  < 0.01, *** *p*  < 0.001, versus CIA vehicle groups, as determined by a one-way ANOVA with Dunnett test.

**Figure 4 plants-09-01535-f004:**
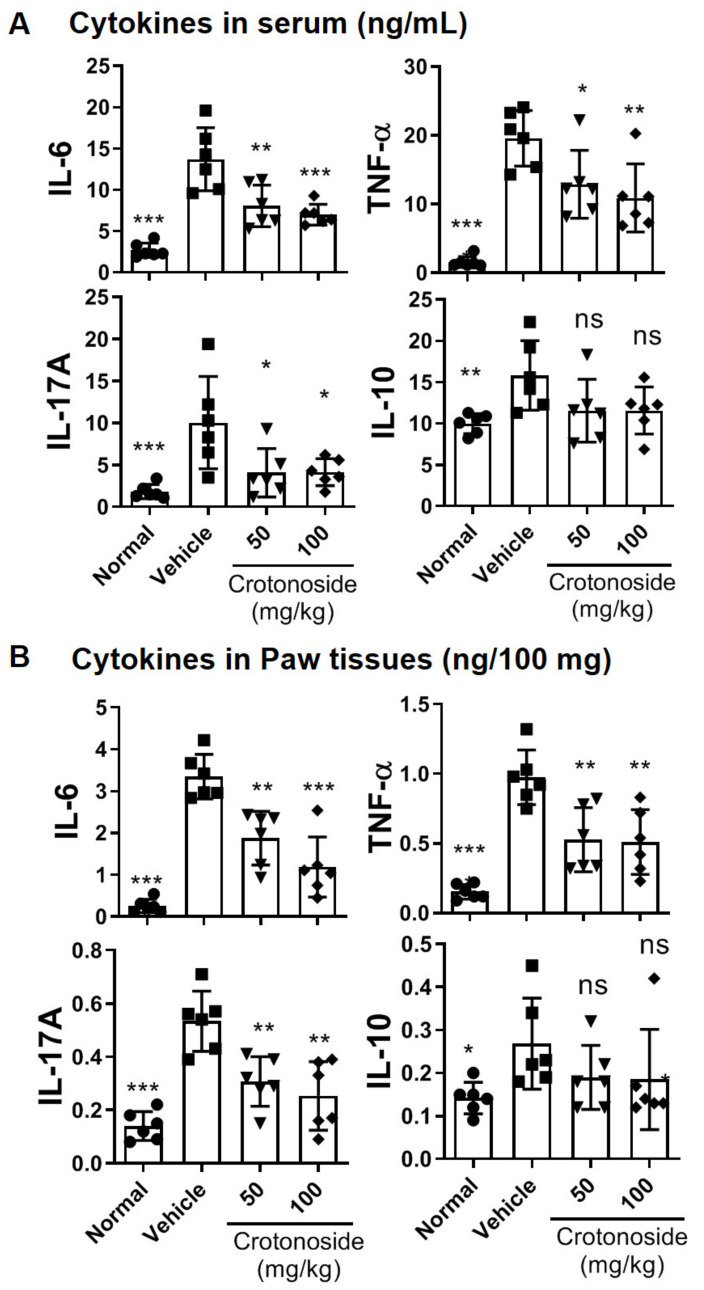
Evaluation of cytokine production in CIA mice (**A**) serum and (**B**) paw tissue homogenate samples under crotonoside treatments. The expression level of proinflammatory cytokines, IL-6, TNF-α, and IL-17A and anti-inflammatory cytokine, IL-10 were determined by ELISA for each tissue sample. Data (*n* = 6 per group) is a representative of three experiments. Bar graphs represent the mean ± SD with six mice per group pooled from two independent experiments. * *p*  < 0.05, ** *p*  < 0.01, *** *p*  < 0.001, versus CIA group, as determined by a one-way ANOVA with Dunnett test.

**Figure 5 plants-09-01535-f005:**
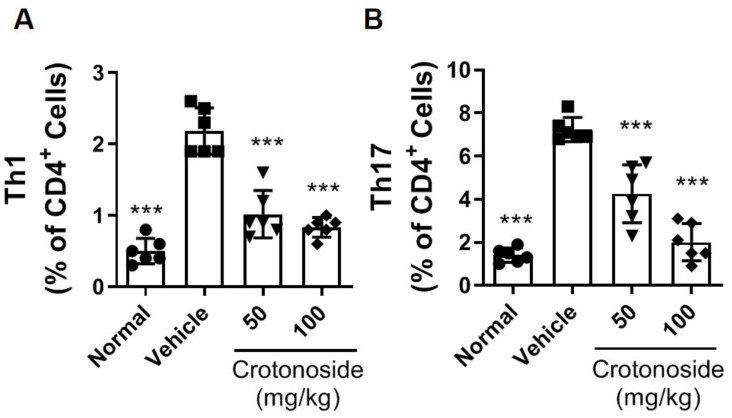
Reduced T cell differentiation from CIA mice by crotonoside. Determination of CII-activated splenic T cell subsets treated with or without crotonoside were stained with anti-CD4 antibody followed by intracellular cytokine staining for IFN-γ, or IL-17A. The percentages of (**A**) Th1 subset (CD4^+^ IFN-γ^+^) and (**B**) Th17 (CD4^+^ IL-17A^+^) were determined by flow cytometry. Bar graphs represent the mean ± SD from two independent experiments where each sample represents a pool of six mice. *** *p* < 0.001, versus CIA/PBS group, as determined by a one-way ANOVA with Dunnett test.

**Figure 6 plants-09-01535-f006:**
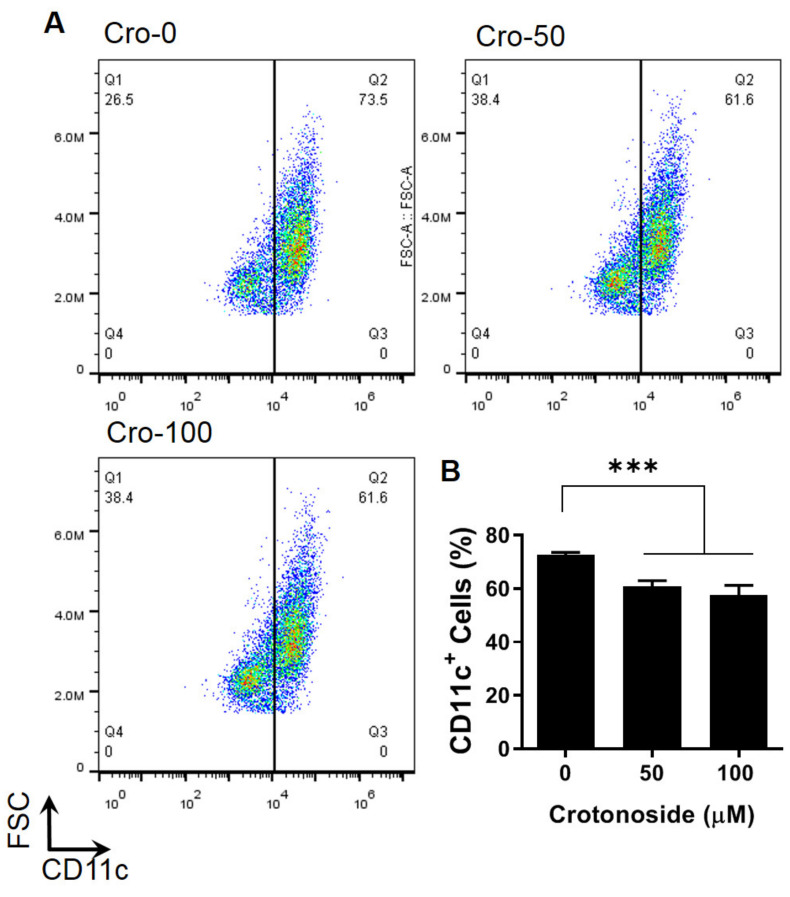
Reduced differentiation of CD11c^+^ bone marrow derived dendritic cells (BMDCs) due to crotonoside treatment. (**A**) The bivariate graphs of BMDCs of CIA mice cultured with the indicated concentration of crotonoside followed by evaluation of CD11c^+^ expressions. (**B**) The mean ± SD of the experimental triplicates are presented in the bar graph. Experiments were repeated three times independently. *** *p* < 0.001 versus 0.1% DMSO-treated mock group, as determined by a one-way ANOVA with Dunnett test.

**Figure 7 plants-09-01535-f007:**
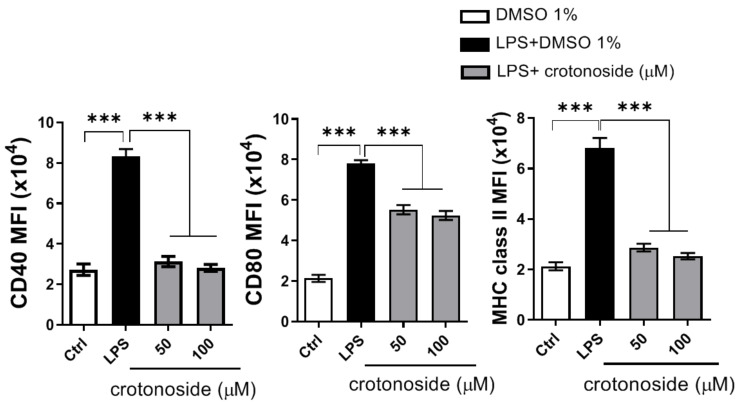
Crotonoside reduced LPS-induced surface co-stimulatory markers of BMDCs. BMDCs from naïve mice were stimulating with LPS and treated with or without crotonoside at indicated concentrations followed by evaluation of co-stimulatory marker expression. Mean fluorescence intensity (MFI) of cellular marker molecule expression was quantified after gating on CD11c^+^ cells and representative data from one of three experiments with similar results was graphed as mean ± SD of triplicate wells. *** *p* < 0.001 versus LPS+DMSO mock control group, as determined by a one-way ANOVA with Dunnett test.

**Figure 8 plants-09-01535-f008:**
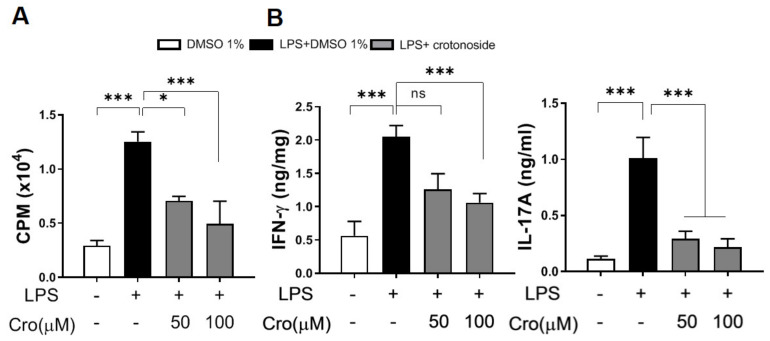
Crotonoside inhibited Ag-specific CD4^+^ T-cell proliferation and cytokine production of LPS-stimulated BMDCs. BMDCs which were activated by OVA and LPS were co-cultured with OT-II-specific T cells followed by determination of (**A**) T cell proliferation via 3H-TdR incorporation assay and (**B**) IFN-γ and IL-17A productions secreted from OT-II CD4^+^ T cells via sandwich ELISA. Values represented as mean ± SD in triplicate and bar graphs represent one of three independent experiments. * *p* < 0.05, *** *p* < 0.001 versus the group of LPS+0.1% DMSO mock control, as determined by one-way ANOVA with Dunnett test.

**Figure 9 plants-09-01535-f009:**
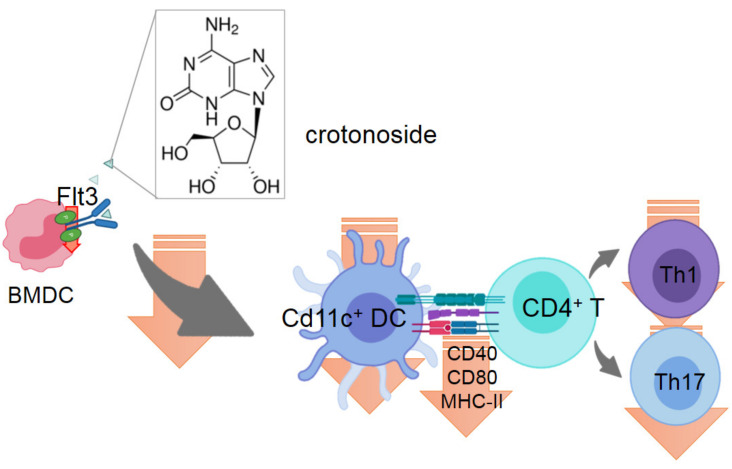
Proposed model of mechanisms exploited by crotonoside to alleviate the CIA in the mouse model. Upon administration of crotonoside, the DC maturation and differentiation were reduced and the capability of priming T cells due to lower CD40, CD80, and MHC-II expressions. Subsequently, the T cell differentiation into pathological Th1 and Th17 were repressed.

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
