# Peer review of "Alleviation of Collagen-Induced Arthritis by Crotonoside through Modulation of Dendritic Cell Differentiation and Activation"

_plants, 2020, doi:10.3390/plants9111535_

Round 1

Reviewer 1 Report

The paper is well written, the prospective study is well presented.

Data are well presented in figures and in text.

Overall, this is a well conducted and presented study.

The minor change is required before publication.

Materials and Methods section:

2.6 Cytokine measurement in vitro and in vivo

If possible, please insert the methods used for preparation of bone marrow-derived DCs, which appeared in 2.9 Flow cytometry------

Reviewer 2 Report

This study is interesting and the manuscript is well written. The conclusion is supported by their experiments. However, there are several issues that require further clarification:

  1. The source of MTX and crotonoside should be clarified.
  2. Figure 1, the meaning of ‘**’in reports of statistical significance is missing. Additionally, it would be better to include both splenic DC and synovil DC in the subtitle of the figure 1 legend.
  3. The rationale of using splenic DC and BMDC for indicated experiments should be clarified. Why authors did not use peripheral blood DC for those experiments?
  4. Figure 3, it would be better to present MTX positive control as Figure 1 and 2 did.
  5. Since RA is characterized by a Th1/Th17 imbalance, the ratio of Th1/Th17 would be better than their individual absolute value to describe the effect of crotonoside on RA.

Reviewer 3 Report

The manuscript describes interesting data on the anti-arthritis activity of crotonoside in an arthritis mouse model. The results suggest consistent anti-inflammatory activity of crotonoside treatment in arthritis. The report is well presented and organized in terms of methodology and experimental design with experiments performed both in vivo and in vitro.

Nevertheless, some comments/criticisms should be addressed, and, consequently, a minor revision of the manuscript is necessary.

In particular,

- the Introduction Section is too short. Crotonoside is not introduced as a Flt3 inhibitor, as it has been in detail described in the Discussion section (from line 384 to line 404). I could suggest to move part of the test regarding Flt3 from Discussion to the Introduction section.

- in the Material and Methods section (lines 212-214), the dilution of the primary antibodies is missing.

- in the Results at the line 341 is reported: “in a dose dependent manner (Fig.7)”. I do not agree, as the differences between 50 and 100 uM crotonoside seem to be not significant among them.

- in the Discussion section (line 421) is reported: “consistent with the reduced phosphorylation of Erk in Supp fig 3”. I do not agree because, although at 90 min there is a different level of Erk phosphorylation, at the Erk activation peak (time points: 30 and 60 min), differences obtained are not statistically significant.

After these corrections, the manuscript will be suitable for publication

Reviewer 4 Report

The manuscript reports the evaluation of the anti-arthritic activity of crotonoside, a guanosine analog originally isolated from Croton tiglium, and its exploration of its immunomodulatory properties in alleviating the severity of arthritic symptoms. The results achieved suggest that crotonoside treatments remarkably improved clinical arthritic symptoms in CIA mouse model and suppressed expression of co-stimulatory molecules, CD40, CD80 and MHC class II, on CD11c + DCs from the CIA mouse spleens. The outcome of this study highlights the immunotherapeutic role of crotonoside in alleviation of CIA via modulation of pathogenic DCs.

The work carried out by the authors was conducted with adequate means. The paper is in most of its parts readable.

- English requires some minor checking.

- 2.2. Animals. Why eight-week-old and why only female mice?
